# High latitude Southern Hemisphere fire history during the mid-late Holocene (750- 6000 yr BP)

Dario Battistel[1,2], Natalie M. Kehrwald[3], Piero Zennaro[1], Giuseppe Pellegrino[1], Elena Barbaro[1,2], Roberta Zangrando[2], Xanthi X. Pedeli[1], Cristiano Varin[1], Andrea Spolaor[2], Paul T. Vallelonga[4], Andrea Gambaro[1,2], Carlo Barbante[1,2]

[1]Department of Environmental Science, Informatics and Statistics, University Ca' Foscari of Venice, Via Torino 155, 30170 Mestre Venezia, Venice, Italy
[2]Institute for the Dynamics of Environmental Processes – CNR, University Ca' Foscari of Venice, Via Torino 155, 30170 Mestre Venezia, Venice, Italy
[3]U.S. Geological Survey, Geosciences and Environmental Change Science Center, Denver, CO 80225, USA
[4]Centre for Ice and Climate, Niels Bohr Institute, University of Copenhagen, Juliane Maries Vej 30, 2100 Copenhagen, Denmark.

*Correspondence to*: Dario Battistel (dario.battistel@unive.it)

**Abstract.** We determined the specific biomass burning biomarker levoglucosan and potassium in ice core from the TALos Dome Ice CorE drilling project (TALDICE) during the mid-late Holocene (750-6000 yr BP). The levoglucosan record is characterized by a long-term increase with higher rates starting at ~4000 yr BP and higher peaks between 1500 and 2500 yr BP. The anomalous increase in levoglucosan centred at ~2000 yr BP is consistent with other Antarctic biomass burning records. Comparisons between levoglucosan and the biomass burning potassium ($K_{bb}$) suggest that potassium can increase the information obtained from a single fire proxy, although $K_{bb}$ is often best used in conjunction with other biomass burning markers. Multiple atmospheric phenomena affect the coastal Antarctic Talos Dome drilling site, where the Southern Annual Mode (SAM) is the most prominent as the Southern Annual Mode Index ($SAM_A$) correlates with stable isotopes in precipitation throughout the most recent 1000 years of the ice core. If this connection remains throughout the mid-late Holocene, then our results demonstrate the changes in biomass burning, rather than changes in atmospheric transport, are the major influence on the TALDICE levoglucosan record. Comparisons with charcoal syntheses help evaluate fire sources, showing a greater contribution from southern South American fires rather than from Australian biomass burning. The levoglucosan peak centred at ~2000 yr BP occurs during a cool period throughout the southern hemisphere, yet during a time of increased fire activity in both northern and southern Patagonia. This peak in biomass burning is influenced by increased vegetation in southern South America from a preceding humid period, where the vegetation desiccated during the following cool, dry period. The Talos Dome ice core record from 6000 yr BP to ~750 yr BP currently does not provide clear evidence that the fire record may be strongly affected by anthropogenic activities during the mid-late Holocene, although we cannot exclude at least a partial influence.

**1 Introduction**

Fire and climate reciprocally influence one another. Biomass burning affects the chemical composition of the atmosphere, the global carbon cycle and the radiative balance due to the emission of greenhouse gasses (carbon dioxide, carbon monoxide, methane and nitrous oxide) as well as aerosols (Andreae and Merlet, 2001; Akagi et al., 2011; Bowman et al., 2009; Keywood et al., 2013; Galanter et al., 2000; van der Werf et al., 2004; Harrison et al., 2010). Temperature, precipitation and atmospheric carbon dioxide ($CO_2$) control both fuel productivity and flammability where biomass growth, fire ignition and spread are favoured by the oscillation of wet and dry conditions (Westerling et al.,2006; Daniau et al., 2010; Marlon et al., 2013; Pyne, 2001; Rollins et al., 2002). Spatial variability in climate and the resulting fire-vegetation-climate interactions further complicate fire dynamics (Lynch et al., 2004). For example, Holocene fire records from the same region may not be synchronous over centennial to millennial timescales (Brunelle and Whitlock, 2003). Furthermore, regional anthropogenic deforestation for the creation of open spaces for croplands and herding adds another variable to the fire-climate system. Anthropogenic impacts complicate fire dynamics, as humans are able to both provoke and extinguish fires (Clare-Smith et al, 2016, Zohary et al., 2012; Tauger, 2013; Ruddiman, 2003; Marlon et al., 2008; Power et al., 2008, Chuvieco et al., 2008; Archibal et al., 2009). The early anthropogenic hypothesis proposed by Ruddiman (2003) is still debated in terms of the scale of the effect of early agriculture on the global climate system, but there is no doubt that land use changes affect climate at regional scales (Broeker and Stocker, 2006; Joos et al., 2004; Singarayer et al., 2011; Mitchell et al., 2013, Kaplan et al., 2009 and 2011).

Charcoal is an established proxy for reconstructing past fire regimes from sedimentary archives. Distinguishing microscopic (< 100 µm) from macroscopic (>200 µm) charcoal allows reconstructing fire history at regional and local scales, respectively (Clark et al., 1996; Marlon et al., 2013; Carcaillet et al., 2002; Whitlock et al.,2007). The Global Charcoal Database (GCD) (Power et al. 2010, https://www.paleofire.org/) provides a useful dataset for research sedimentary records of fire (Blarquez et al., 2014; Marlon et al., 2008 and 2013; Power et al., 2008 and 2013; Daniau et al., 2012; Mooney et al., 2011; Vanniere et al., 2011) and the paleofire R package creates customized charcoal syntheses (https://github.com/paleofire/GCD, Blarquez et al., 2014). Although the recent Global Charcoal Database version 3 (GCDv3) compiles more than 700 charcoal records (Marlon et al. 2016), the sites included in GCD are not homogenously distributed, due in large part to the location of lakes, leading to geographical over- or underrepresentation. The atmospheric transport of macrocharcoal only extends a few kilometers (Carcaillet et al 2001) and therefore charcoal records cannot encompass some areas such as deserts or polar regions.

Ice cores complement charcoal syntheses as they can reconstruct fire histories at regional to semi-hemispheric scales (Legrand et al. 1992). Fire proxies in ice cores include inorganic chemicals such as ammonium, nitrate and potassium (Legrand et al. 2016), black carbon (McConnel, 2007) and organic compounds such as formate, oxalate, phenolic compounds and levoglucosan (Legrand et al., 2016; Zangrando et al., 2013 and 2016). These proxies originate from biomass burning, but not all of these materials are exclusively emitted by combustion. Several organic compounds are strongly related to the

type of fuel where, for example, conifer combustion produces dehydroabietic acid (Fine et al. 2002). However, other proxies such as oxalate, phenolic compounds and potassium are not specific indicators of biomass burning. Emission factors and lifetimes differ between proxies as well as atmospheric transport between the source region and the drilling site may influence individual records (Rubino et al.
5   2016).

Levoglucosan (1,6- anhydro β-D-glucopyranose) is a source specific proxy for fire that is produced by cellulose and emitted by biomass burning, with a maximum yield centered around ~250°C (Kuo et al., 2008). The atmospheric lifetime of levoglucosan is actively debated with estimates ranging between a
few days (Hoffman et al., 2010; Hennigan et al., 2010) to a few weeks (Bai et al., 2013; Slade and Knopf, 2013). Ice cores levoglucosan reconstructions currently provide biomass burning histories from Northern Hemisphere locations. Levoglucosan records extending back 15,000 years reconstruct high northern latitude fire history in Greenland (Zennaro et al., 2015). Mountain glacier levoglucosan records depict regional fire histories in Kamchatka and the Tibetan Plateau (Kawamura et al. 2012; Yao et al.
2013). However, Southern Hemisphere levoglucosan records still leave much to be explored. Antarctic levoglucosan records will likely differ from Arctic records due to the substantially smaller land masses surrounding near the ice sheet and the long distances required for atmospheric transport of biomass burning material. The southernmost tip of Patagonia, the closest continental landmass to Antarctica, only extends to ~55°S, whereas the Arctic contains the largest land masses in the world. Although long-
term levoglucosan records have never been reported for Antarctic ice cores, several studies determine other biomass burning by-products (i.e. secondary organic aerosol and black carbon) during the last few decades and centuries (Wolff and Cachier, 1998, Fiebig et al. 2009, Hara et al. 2010, Hu et al. 2013, Weller et al. 2013, Pasteris et al. 2014) as well as during the Holocene (Arienzo et al. 2017).

Hemispheric fire history since the Last Glacial Maximum, derived from marine records (Daniau et al. 2012), demonstrates relatively high fire activity during the Holocene where biomass burning is more pronounced at high latitudes. The northern and southern hemisphere fire histories substantially differ from one another (Daniau et al. 2012) where the southern hemisphere fire history is characterized by a widespread spatial heterogeneity (Power et al., 2008). From the mid-late Holocene onward, sea levels
approached near-modern levels, and most regional Southern Hemisphere glacial ice had melted. The increase in human population in Australasia and South America and the associated shifting vegetation types and/or the strengthening of the El-Niño Southern Oscillation (ENSO) activity explain the heterogeneity of fire patterns since 3000 y BP (Power et al 2008, Whitlock et al. 2007).

We aim to reconstruct Southern Hemisphere fire history by determining levoglucosan concentration in an ice core obtained from the TALos Dome Ice CorE drilling project (TALDICE) during the mid-late Holocene (750-6000 yr BP). Talos Dome (159°11'E, 72°49'S, 2315 m a.s.l., Fig.1) is located in the South Pacific/Ross Sea sector of the East Antarctic Plateau (www.taldice.org). The relatively high snow accumulation rates (80 kg m$^{-2}$ yr$^{-1}$, average 2004-1259 AD; Stenni et al. 2002) enables accurate dating
of the core (Buiron et al., 2011; Veres et al. 2013) and high-resolution climate analyses during the Holocene (Albani et al., 2012a; Delmonte et al., 2013). We also analyze potassium in order to provide a more complete biomass burning record through a comparison with another fire proxy. Charcoal

syntheses provide information of biomass burning source regions and changing fire activity. These fire and climate records can provide a synthesized history of high latitude Southern Hemisphere biomass burning in a relatively stable climate regime that is subject to an increasing human influence.

## 2 Methods

**2.1 Ice core samples and analysis**

We analyzed 266 ice core samples from the TALDICE (TD) ice core, covering the depth interval between 90 and 403 m. Each sample represents the uppermost 15 cm of a 1 m ice core section. These sections were the ice remaining after other analyses performed by Italian TALDICE researchers. The samples are therefore discontinuous and equidistant in space but not in time. The samples thus have the

potential to miss fire peaks due to their discontinuous nature. The ice core samples were stored at Ca' Foscari University of Venice at -20°C until analysis. Before the analysis, the ice samples were decontaminated by washing the outermost section three times using ultrapure water (ELGA LabWater, Marlow, UK) in an ISO 5 clean room in order to remove possible drilling fluid residuals or impurities. During the washing, the samples were held using PTFE tongs (Nalgene Corporation, Rochester, NY)

previously immersed for two days in 5% and 1% $HNO_3$ solutions and rinsed with ultrapure water. The same procedure was used for decontaminating LDPE bottles and vials. Ice samples were melted at room temperature in 125 mL LDPE bottles and then transferred to 15 mL LDPE bottles and stored at -20°C in triple polyethylene bags. These precautions were necessary for minimizing any possible contamination of the samples.

Levoglucosan analysis was performed using a method specifically developed for the analysis of polar ice samples (Gambaro et al. 2008 and as modified in Zennaro et al., 2014 and 2015) based on liquid chromatography coupled with a triple quadrupole mass spectrometer (HPLC/(-)ESI-MS/MS). Briefly, we injected 300 µL of the sample in a HPLC system (Agilent 1100, Waldbronn, Germany).

Chromatographic separation was obtained using a C-18 Synergy Hydro column (4.6 mm i.d., 50 mm length, 4 µm size particles; Phenemenex, Torrance, CA) and an autosampler was equipped with a LOOP Multidraw 44 Upgrade Kit G1313 – 68711. The samples were eluted with methanol (ultragradient, H411, Romil Ltd. Cambridge, U.K) and ultrapure water (18.2 MΩ, TOC 1 ppb, PURELAB Pulse and PURELAB Flex, Elga). The mass spectrometer was an API 4000 (Applied

Biosystem/MDS SCIEX, Toronto, Ontario, Canada) equipped with an ion spray source ((-)ESI) Turbo V operating in negative polarity. Mass/charge (m/z) ratios used for the quantification were 161 and 113 for levoglucosan and 167 and 118 for $^{13}C$-labelled levoglucosan.

This method limits pre-analytical procedures that may affect the quantitative determination and the

sample contamination and it allows analyzing levoglucosan at levels of a few pg mL$^{-1}$. We quantified the results using isotopic dilution ($^{13}C_6$-labelled levoglucosan as an internal standard; Cambridge Isotope Laboratories Inc. (Andover, MA)) and instrumental response factors were determined prior, during and after each set of analyses in order to evaluate eventual deviation from the instrumental response. The instrumental limit of quantification (LoQ) for levoglucosan was 4 pg mL$^{-1}$, determined

following the analytical method reported in Gambaro et al. 2008. From an analytical point of view, a reliable procedural LoQ is difficult to determine in this case, due to the lack of a suitable aqueous matrix. The glacier water often contains lower concentrations of levoglucosan than the ultra-pure laboratory water, thereby complicating obtaining a true LoQ. However, as this method only requires as a few pre-analytical procedures, where these steps are always performed in a dedicated clean room, we therefore use the instrumental values as the LoQ.

The analytical methods for the analysis of Na, Fe and K are previously reported (Vallelonga et al. 2013). Melted samples were acidified to pH 1 using sub-boiling distilled $HNO_3$ (Romil, Cambridge, UK) and analyzed at least 24 hours later by Inductively Coupled Plasma Sector Field Mass Spectrometry (ICP-SFMS; FinniganTM ELEMENT2, Thermo Fisher Scientific Inc., Bremen, Germany) coupled to an APEX Q desolvating introduction unit (Elemental Scientific, Omaha, NE, USA) in low ($^{23}Na$), medium ($^{56}Fe$) and high ($^{39}K$) resolution modes. The detection limits, defined as three times the standard deviation of the instrumental blank, and analytical precisions were (0.06 ng mL$^{-1}$, 3%), (0.03 ng mL$^{-1}$, 16%) and (0.2 ng mL$^{-1}$, 3%) for Na, Fe and K, respectively. Levoglucosan, K, Na and Fe concentrations and their associated errors are reported in the Supporting Material (see file rawdata.xls).

**2.2 Dating and data treatment**

The ice cores sample ages were calculated using the AICC2012 age scale (Bazin et al. 2013; Veres et al. 2013). The depth-age model and associated errors are also reported in the Supplementary Material (rawdata.xls). Using this depth-age model, we normalized levoglucosan and biomass-burning potassium ($K_{bb}$) concentrations into flux values by dividing the concentrations by the accumulation rate for each corresponding ice section (Buiron et al. 2011).

In addition to levoglucosan, other proxies also depict biomass burning in ice cores. Potassium is one of the initial biomass burning proxies (Cachier et al. 1991). Some organic matter emits potassium when burned, where the potassium is then incorporated into atmospheric air masses. Unlike levoglucosan, potassium is not a source-specific proxy. The primary sources of potassium in aerosols are soil, sea-salt and biomass burning smoke where marine and terrestrial input affects the potassium contribution from fire plumes (Legrand and de Angelis 1996; Savarino and Legrand 1998). Several authors propose the following formula to obtain biomass burning potassium ($K_{bb}$) by correcting for the terrestrial and marine contributions to the total potassium concentrations (Begum et al., 2006; Cahill et al., 1980):

$$[K_{bb}] = [K] - 0.6[Fe] - 0.038[Na], \tag{1}$$

where [K], [Fe] and [Na] are the potassium, iron and sodium concentrations in the sample, respectively. Legrand and de Angelis (1996) propose using calcium instead of iron to account for the terrestrial contribution of potassium in Greenland firn cores ($[K^+]/[Ca^{2+}] = 0.04$). Due to the substantial local influence on $Ca^{2+}$ concentrations (see section 3.2 for a more detailed discussion) we evaluate $K_{bb}$ using Eq.(1).

Charcoal syntheses were performed by using the paleofire R package associated with the Global Charcoal Database by extracting a 20-year bin time window (consistent with the time period covered by each levoglucosan sample) from 6000 y BP to present. Detection and separation of anomalous intense peaks from the levoglucosan and $K_{bb}$ series were based on Cook's distance (Cook, 1977), after generalized additive modeling (GAM) of the relationship between each series and age using the R package mgcv (Wood, 2006). Preliminary data analysis revealed that levoglucosan records were highly skewed, and we therefore log-transformed all data to reduce skewness before any further analysis. Potential associations between levoglucosan and $K_{bb}$, as well as atmospheric transport indicators, fire sources and climate parameters, were analyzed using a slot correlation with Gaussian kernel, in order to account for the irregularly sampled data (Rehfeld et al., 2011). In order to investigate the statistical significance of the resulting correlation estimates (r), we constructed 95% bias corrected and accelerated (BCa) confidence intervals based on 5000 bootstrap samples. Calculations were made in R (R Core Team, 2017) using the boot package (Canty and Ripley, 2017; Davison and Hinkley, 1997).

## 3 Results and Discussion

### 3.1 Fire tracers in the Talos Dome ice core

In the 266 TD ice core samples analyzed in this work, levoglucosan concentrations ranged from 4 (corresponding to the LoQ) to 1100 pg mL$^{-1}$. However, in 68 of the 266 samples the levoglucosan concentrations were below the LoQ. The highest concentrations were lower, but of the same order of magnitude, than the major levoglucosan peaks detected in the NEEM (Greenland) ice core (185-1767 pg mL$^{-1}$) during the late Holocene (Zennaro et al., 2014) and in the Ushkovsky (Kamchatka Peninsula) ice cap (10-5000 pg mL$^{-1}$) during the last three centuries (Kawamura et al., 2012). Levoglucosan concentrations from these remote regions are significantly lower than in environmental archives closer to more human-influenced areas such as in the Muztagh Ata and Tanggula ice cores where concentrations were 10-718 ng mL$^{-1}$ and 10-93 ng mL$^{-1}$, respectively (Yao et al., 2013).

The complete mid-late Holocene Talos Dome levoglucosan record is shown in Fig. 2B. Levoglucosan spikes potentially indicate extreme fire events, more frequent fire activity during the short temporal interval, and/or more efficient atmospheric transport during the fire event. These anomalous intense peaks significantly affect the temporal trend of the record even when adopting different smoothing approaches (Zennaro et al., 2014). In order to better identify which individual peaks could influence the entire dataset, we used Cook's distance to indicate outliers (n=9) which resulted in peaks with values greater than ~ 16 µg m$^{-2}$ yr$^{-1}$ (Fig. 2B; Table 1). We therefore separate these intense peaks from the longer-term fire history (Fig.2C).

The levoglucosan flux is lowest during the oldest part of the record (between 6000 and 4000 yr BP), while a significant long-term increase occurs after approximately 4000-3500 yr BP (Fig. 2A-1). Levoglucosan increases between ~2000 and ~2250 yr BP, and with a major increase between ~1250 to ~800 yr BP. The timing of the individual levoglucosan spikes (Table 1) roughly occur between 2200-1800 and 1200-900 yr BP, corresponding with the major long-term increases. Thus, these extreme peaks

should be considered as part of a more widespread period when levoglucosan was generally higher in the Talos Dome Mid-Late Holocene sequence rather than isolated fire events.

Attributing these individual spikes only to megafire events, however, may oversimplify their source. OH• radicals, that can occur in the atmosphere in both the gas and aqueous phase, are able to oxidize levoglucosan, leading to degradation during transport (Hennigan et al., 2010; Hoffman et al., 2010). The extent of this degradation is currently unknown as recent studies also propose that levoglucosan can exist in the atmosphere up to 26 days (Bai et al. 2013). The atmospheric stability of levoglucosan and atmospheric transport dynamics both influence fire signals in polar ice (Zennaro et al., 2014; Legrand et al., 2016). Although large amounts of levoglucosan are emitted during biomass burning and are quantifiable even at remote distances from the source (Zennaro et al., 2014 and 2015, Kehrwald et al., 2012), the detected levoglucosan concentrations may be affected by a combination of transport and degradation, thus making the signal over or under-representative of the initial individual fire event. Conditions such as varying precipitation amounts, dry versus wet deposition, and the distance from the fire source can modulate the levoglucosan signal, thus resulting in reducing the effects of megafire events or amplifying modest fires. However, the stability of levoglucosan once trapped in the ice may be less significant. Levoglucosan profiles in polar ice cores do not result from simple degradation curves, supporting the assumption of the stability of this proxy in ice cores (Zennaro et al., 2014 and 2015, Kehrwald et al., 2012). Having multiple proxies that identify the phenomenon of interest (i.e. biomass burning) provides a more robust signal as the strengths of each proxy can help balance the weaknesses of the others. We therefore also investigate $K_{bb}$ to compare the levoglucosan results with an independent fire proxy (Fig. 2).

We applied Cook's distance to the $K_{bb}$ profile in the same manner as for the levoglucosan profile in order to determine anomalous peaks (n=9, Fig. 2D; Table 1) and to then examine the long-term trend without the influence of these peaks (Fig. 2E). The $K_{bb}$ profile significantly increased at ~4000 yr BP, with other high values at 2000-2500, 2750-3000 and 3500-3750 yr BP, followed by a further increase between 1250-750 yr BP (Fig. 2E). The $K_{bb}$ profile only partially agrees with levoglucosan. For example, $K_{bb}$ increases between ~3000 and ~4000 yr BP where this peak is not consistent with the levoglucosan record. These differences may be due to the non-specificity of potassium for biomass burning where local to regional terrestrial sources may influence the $K_{bb}$, trend, leading to differences with the levoglucosan signal. The multi-source aspect of potassium remains a critical issue when reconstructing fire histories using this proxy.

The outlying values of $K_{bb}$ (Table 1) mainly occur after 2200 yr BP and are present in two distinct periods between 900-1200 yr BP (group I) and 1900-2200 yr BP (group III). These periods are similar to the time intervals with outlying values in the levoglucosan record. Levoglucosan and $K_{bb}$ spikes in group I perfectly coincide at 1180 and 1193 yr BP, including their relative amounts after applying a minimax transformation. In group I, only sample TD108 does not have a direct match between $K_{bb}$ and levoglucosan peaks. In the other samples included in group I, a direct comparison is not possible because the samples were not simultaneously analyzed for levoglucosan and $K_{bb}$. In contrast, between 1600-1900 yr BP (group II), the levoglucosan and $K_{bb}$ peaks do not correspond, as only levoglucosan

was detected during this time period as $K_{bb}$ consistently remained below the limit of detection. In Group III (between 1900-2200 yr BP) the peaks perfectly correspond at 1920 yr BP, while in the short period between 2160-2200 yr BP levoglucosan and $K_{bb}$ peaks were close to one another, although not coincident, and are offset by approximately 15-25 years. Considering that we analyzed the same ice samples, a possible uncertainty in chronology is not sufficient to explain this discrepancy. It is possible that potential differences in migration of $K^+$ ions and levoglucosan within the ice matrix and/or selective depositional/ post-depositional processes result in this offset between the two proxies, although migration of soluble inorganic ions appears to be limited (Kreutz et al. 1998).

The individual levoglucosan and $K_{bb}$ spikes occur during the same time windows, where these time intervals also correspond to high long-term values, suggesting periods of increased fire activity. The relationship between levoglucosan (log-transformed) and $K_{bb}$ profiles was estimated using the slotting correlation approach. Data indicate a positive and significant correlation between the two variables, both including (r (95% CI) =0.264 (0.141, 0.536)) and excluding outliers (r (95% CI) =0.394 (0.280, 0.563)). This comparison between the specific biomarker levoglucosan and the multi-source $K_{bb}$ suggests that $K_{bb}$ can increase the information obtained from a single fire proxy, but that $K_{bb}$ is often best used in conjunction with other biomass burning markers. In the following discussion, we focus on the levoglucosan record for reconstructing fire activity during the mid-late Holocene.

The attribution of the levoglucosan spikes to individual large fire events is difficult to assess as atmospheric transport and stability may alter the signal, resulting in an amplification of modest fires. A similar situation occurs for individual peaks in charcoal records as these spikes can either be intense local fire events or can result from an increase in short-term transport. For example, the high charcoal signals observed in Laguna Padre Laguna (Argentina) between ~1500 and ~2000 yr BP and at Laguna Zeta (Argentina) between ~2000 and ~2500 yr BP (Iglesias and Whitlock, 2014) are consistent with the spikes observed in the levoglucosan record, as well as intense fire episodes recorded between ~2100 and ~2300 yr BP in the Wingecarribee Swamp (Southern Australia (de Montford, (2008); ID site=857 in the GCD) and at 2160 yr BP in Eweburn Bog (New Zealand, (Ogden et al. 1998)); ID site=441 in the GCD). A possible correspondence with levoglucosan and these individual charcoal spikes is only speculative. Comparing trends in long-term fire activity, as identified by smoothed levoglucosan records and charcoal syntheses, is more indicative of changes in biomass burning as opposed to comparing individual, likely localized, events. We therefore only discuss biomass burning trends rather than individual spikes from this point forward.

### 3.2 Atmospheric transport

Unlike central East Antarctic Plateau locations (i.e. Vostok (Petit et al., 1999) and EPICA Dome C (EDC) (Lambert et al., 2008)), Talos Dome is significantly influenced by dust of local origin. Talos Dome contains dust grains larger than 5 µm (diameter) that are absent in the EDC core (Delmonte et al. 2010a). The finer dust fraction (1-5 µm) at Talos Dome is considered to be a mixture of local and remote sources, while this same fraction in EDC is due to long-range transport where the net dust amount is substantially less than at Talos Dome. However, a clear declining trend in dust flux since

~8000 y BP was evident in both ice core records (Albani et al., 2012a; Delmonte et al., 2005). The progressive opening of the Ross Sea embayment from the Early Holocene onward increased the relative contribution of regional dust (Hall et al., 2009; Stenni et al., 2011; Masson-Delmotte et al., 2011; Albani et al., 2012 a).

The composition of mineral dust provides useful indications of the original source and subsequent atmospheric transport. The isotopic composition of dust in central East Antarctica suggests that southern South America was the primary dust source during glacial periods, with possible contributions from Australia (Delmonte et al., 2010b; Grousset et al., 1992; Basile et al., 1997; Delmonte et al., 2008; Gabrielli et al., 2010; Vallelonga et al., 2010). During the Holocene, the considerable contribution of dust from proximal ice-free areas of northern Victoria Land, Antarctica, located at similar altitude with respect to the drilling site, (Delmonte et al., 2010a) renders the isotopic analysis of Talos Dome dust unreliable. Nevertheless, the $^{87}Sr/^{86}Sr$ vs $^{143}Nd/^{144}Nd$ isotopic signature of mineral dust in other East Antarctica sites (Vostok and Dome C) prompted Revel-Rolland (2006) to consider a possible Eastern Australian provenience. Although the southern South American source was reduced with respect to glacial periods, several authors proposed that the South American provenance also persisted during the interglacials (Holocene and MIS-5), but where the source shifted to lower latitude regions (Delmonte et al., 2008; Gaiero et al., 2007). This hypothesis is also supported by the lithium enrichment factors from the EDC ice core that are similar to continental ice cores from Bolivia (Siggaard-Anderses et al., 2007; Correia et al., 2003). However, a relative weakening of the South American source during interglacials cannot be excluded (Delmonte et al., 2007 and 2008; Gaiero et al., 2007) with a feasible combination of both Australian and South American contributions during the Holocene.

We cannot stress strongly enough that the source regions and atmospheric transport of mineral dust and organic materials in smoke plumes may not be the same. Comparisons between Talos Dome levoglucosan (log-transformed) and mineral dust <5 µm (log-transformed data from Albani et al. 2012) provide a statistically significant negative correlation (r (95% CI) = -0.215 (-0.365, -0.103)). These results suggest that at least part of the levoglucosan record at Talos Dome may be affected by a different circulation pathway than that of the mineral dust.

**3.3 Biomass burning sources**

The levoglucosan profile likely reflects fire activity occurring in the nearby continents, as Antarctica contains essentially no biomass available for burning. Although charcoal is a local proxy for fire (Whitlock and Larsen 2002; Clark et al., 1996 and 1988), synthesizing individual records from the Global Charcoal Database (Marlon et al. 2016; Blarquez et al. 2014) is a useful tool for reconstructing fire history across regions (Marlon et al. 2013 and 2016). We compiled charcoal syntheses from the following potential source areas of the fire signal recorded in the Talos Dome ice core: (i) East Australia (including New Zealand) between 90° and 30°S (latitude) and between 160° and 180° E (longitude) (ii) southern South America between 90° and 30°S (latitude) and between 60° and 80° W (longitude) and (iii) a synthesis of the Southern Hemisphere covering areas between 90° and 30° S (Fig.3). The number of charcoal records varies by region where the East Australia and southern South America composite

records were obtained by combining n=131 and n=25 sites, respectively. In order to compare the levoglucosan record with the charcoal syntheses, we applied a Box-Cox transformation to the records (Power et al. 2008). As is evident from Fig.3, the levoglucosan increase observed at ~3500 yr BP coincides with the same increase in southern South America and Southern Hemisphere charcoal syntheses. Conversely, the levoglucosan increase centered at ~2000 yr BP, which also includes more frequent outliers is not supported by any charcoal records. The correlation analysis of Talos Dome levoglucosan (log-transformed; outliers excluded) and charcoal records produces a statistically significant positive correlation with southern South America (r (95% CI) =0.333 (0.242, 0.438)) but not in East Australia (r (95% CI) =0.011 (-0.078, 0.095)). Levoglucosan does positively correlate with the total SH charcoal synthesis (r (95% CI) =0.225 (0.138,0.317)) over the investigated time period of 750-6000 yr BP.

Talos Dome Levoglucosan and southern South America charcoal records (Fig.3) both depict an increasing long-term trend in biomass burning from 4000 to 750 yr BP, although these trends do not perfectly coincide. These records differ both before and after these ~3000 years of similarity. The Talos Dome record begins its major and sustained increase in fire activity around 4500 yr BP, while the southern South American record is decreasing during this time period. The southern South America and East Australian records substantially diverge after 750 yr BP, where the southern South American fires decrease and the East Australian and SH fires both markedly increase. The Talos Dome levoglucosan record does not extend to the present, and so comparisons with this major difference in regional fire sources are not possible.

These results may suggest a higher contribution from southern South American fires rather than Australian biomass burning over centennial to millennial timescales between 4000 to 750 yr BP, although the contribution from East Australia cannot be completely neglected. However, it is also possible that this observation may be due to the higher regional variability in East Australian fire activity (Mooney et al., 2011). During the Mid-Holocene, sites in the far south of South-Eastern Australia showed higher fire activity due to a shift toward more moisture-stressed vegetation, while sites in the Southern Tablelands and east of the Great Dividing Range showed less fire activity with the region supporting relatively moisture-demanding vegetation (Pickett et al., 2004; Mooney et al., 2011). Thus, over a wider spatial scale, these contributions tend to cancel one another. As a result, although the South-Eastern Australia charcoal synthesis is characterized by higher values at about 6000 yr BP, during the Mid-Holocene, only small local increases at ~4000 y BP and ~2500 yr BP were observed. The southern South American fire synthesis is generally more homogenous, although some regional spatial variability between northern and southern Patagonia does exist (Mundo et al. 2017). The southern South American region defined in these charcoal syntheses extends from Patagonia to the Argentinian Pampas, encompassing grassland, steppe, desert and forest vegetation.

### 3.4 Drivers of biomass burning

The iron content in Southern Chile marine sediments reconstructs rainfall changes due to latitudinal shifts of the Southern Westerlies during the Holocene (Lamy et al., 2001; see also Fig.4 in this paper).

This reconstruction is consistent with regional terrestrial paleoclimate data-sets (Villagran et al., 1990; Lamy et al., 2001) and indicates more arid conditions from 7700 to 4000 yr BP, followed by a general increase in rainfall from 4000 yr BP to the present day. This finding is consistent with the hypothesis that fire activity in southern South America is mainly driven by fuel availability (Iglesias and Whitlock, 2014 and Iglesias et al. 2014). However, the main drivers of fire may vary within Patagonia as fire activity in southern Patagonia depends upon fuel desiccation due to droughts rather than the years of wet climate conditions that result in more vegetation, which is an important driver of fire in northern Patagonia (Mundo et al. 2017). The southern South America precipitation record (Lamy et al., 2001) and Talos Dome levoglucosan record demonstrate a relationship between rainfall and fire activity in the mid-late Holocene (Fig. 4). Bivariate analysis indicates a significant negative correlation between levoglucosan (log-transformed; outliers excluded) and iron records (log-transformed) (r (95% CI) =-0.183 (-0.277, -0.083)), suggesting that when arid conditions characterized southern South America, the levoglucosan signal was low and started to increase when the southern westerlies strengthened and shifted equatorward, bringing more humid conditions to the continent. This interpretation is consistent with the increase of more humid conditions in continental records (Iglesias and Whitlock, 2014 and Iglesias et al. 2014) although it cannot be excluded that the changes in the westerly wind belt may have increased, at least in part, the transport of aerosols to Antarctica.

Southern Hemisphere temperature (r (95% CI) =-0.272 (-0.428, -0.147); see also Fig.5) substantially decreases between ~1500 and ~2500 yr BP, where this decrease is not evident in either the solar irradiance or greenhouse gas concentrations (Fig. 4). Carbon dioxide increases fairly steadily from the mid-late Holocene onward in the Taylor Dome ice core (Indermühle et al. 1999; Fig. 4), while the general levoglucosan trend increases beginning ~4000 yr BP, yet includes a major dip in fire activity around ~2800 yr BP (r (95% CI) =0.455 (0.357 0.576)). This difference is to be expected, as $CO_2$ in ice cores represent a globally-mixed signal while levoglucosan in polar ice cores is a regional to semi-hemispheric signal (e.g. Zennaro et al., 2014 and 2015). Strong El Niño Southern Oscillation (ENSO) events and periods affect high latitude Southern Hemisphere climate by increasing mean Southern Hemisphere air temperature (Schneider and Steig, 2008), expanding sea ice east of the Antarctic Peninsula (Mayewski et al., 2017 and references therein), and delivering snowfall to East Antarctica (Roberts et al., 2015). The strengthening of ENSO beginning around ~5000 yr BP increased short-term fluctuations in Patagonian vegetation (Moy et al., 2002; Iglesias et al., 2014) and increased precipitation near northern Patagonia as evidenced by the iron record in marine sediments (Lamy et al., 2001; Lamy et al., 2004). The iron record and changes in ENSO strength (log-transformed) also correlate with the Talos Dome levoglucosan record (log-transformed) (r (95% CI) = 0.172 (0.092, 0.354)), although to a lesser extent than solar radiation and greenhouse gases. ENSO variability has a weak teleconnection with fire activity (Bisiaux et al. 2012, Mundo et al. 2017) where the weakness of this connection can be ascribed to the regional nature of ENSO with a greater ENSO influence on northern rather than southern Patagonia.

Variable and generally wet conditions prevailed in South America after 4000 yr BP, resulting in more available biomass and increased fire events. However, biomass burning increased even more when, at ~

2500 yr BP, Patagonia became drier yet still contained an abundance of vegetation from the previous humid period (Iglesias et al. 2014). This 1000-year peak in levoglucosan (between 2450±100 and 1600±1000 yr BP) occurs at the same time as a local decrease in temperature between 1500 and 2500 yr BP. As the vegetation from the previous humid period would not likely remain for almost 1000 years, and as it is unlikely that decreased temperatures triggered an increase in fire activity, we propose three possible explanations: (i) precipitation-evaporation balance and/or aerosol feedbacks (ii) different atmospheric transport and (iii) influence of insolation.

(i)     Studies of long-term fire activity variations since the Last Glacial Maximum and through the Holocene demonstrate that temperature is the main driver of fire activity over a global scale, while the precipitation-evaporation (P-E) balance is the second-most important factor (Daniau et al 2012). The net effect of both temperature and P-E strongly depend on the initial conditions and fuel availability. Fire does not respond linearly to temperature, and an increase in temperature does not directly result in more fire under warm rather than cold conditions (Daniau et al. 2012). The relation between fire and P-E is unimodal, where an increase in P-E under dry/moist conditions leads to an increase/decrease in fire. This long-term observation contrasts with our record, where an increase in fire activity coincides with a net decrease in temperature. It must be noted, however, that our observations do not represent a global signature and the relatively cold period was limited to only approximately 1000 years in the framework of the current interglacial. Moreover, fires emit both large quantities of $CO_2$ and particulates. While $CO_2$ warms the atmosphere, many aerosols can decrease temperature by shielding the solar radiation that reaches the surface. As a net result, at least over local to regional scales, aerosol emissions lead to a cooling effect, as observed in the Kuwait oil fires, where a reduction in ground-level temperature resulted from the absorption of solar radiation by the smoke (Bakan et al. 1991). However, although aerosol emissions can impact the surface temperature, this hypothesis is very difficult to support for a time-period of ~1000 years and over such a large spatial scale.

(ii)     Stacked SH proxy data between 30 -90° S demonstrate a major temperature decrease between 1500 yr to 2500 yr BP (Figure 4; Marcott et al., 2013). During this time period, the Talos Dome, West Antarctic Ice Sheet Divide (WAIS Divide) and the Roosevelt Island Climate Evolution (RICE) ice core isotope and snow accumulation records positively correlate (Bertler et al., 2018) where decreased temperatures are associated with less accumulation and vice versa. This correlation between temperature and accumulation applies to each site, yet during this time period, the stable isotopes of the RICE ice core in the eastern Ross Sea and Talos Dome in the western Ross Sea have phases where they differ from one another, creating a distinct Ross Sea Dipole. During the time period of increased fire activity in the Talos Dome ice core ~2500 to ~1500 yr BP, the following relationships exist between detrended, normalized stable isotopes in Talos Dome and the RICE ice core: ~2500 to ~2150 yr BP in phase, ~2150 yr BP to ~1750 yr BP out of phase, ~1750 to 1100 yr BP in phase (Bertler et al., 2018). This difference suggests that the air mass trajectories affecting these sites were similar when the records were in phase, but diverged when the ice core records are out of phase. In general, this anti-correlation between RICE and Talos Dome is strongest when the Southern Annual Mode Index ($SAM_A$) is negative (Abram et al., 2014). The SAM is defined as the difference in zonal mean atmospheric pressure between ~40°S and ~65°S (Marshall, 2003). This atmospheric link between mid-latitudes and Antarctica leads to

cooler, drier conditions in East Antarctica and warmer, wetter conditions in West Antarctica when the SAM index is positive (Marshall and Thompson, 2016). A positive SAM is associated with a southward shift of storm tracks over South America, and possibly increased transport of aerosols from South America to Antarctica (Christie et al., 2011; Abram et al., 2014). Over the thousand years of the quantified $SAM_A$, the Talos Dome stable isotopes and $SAM_A$ have remarkably similar patterns and timing where the positive $SAM_A$ coincides with increased temperatures deduced from the ice core stable isotope record (Bertler et al., 2018). The coastal location of Talos Dome is neither entirely influenced by the teleconnections impacting West and East Antarctica (Bertler et al, 2018) and so this strong correspondence between $SAM_A$ and the Talos Dome stable isotopes is surprising. If this connection between $SAM_A$ and stable isotopes in precipitation near Talos Dome remains the same back to 2500 year BP, then the 1000-year peak in levoglucosan (between 2450±100 and 1600±1000 yr BP) encompasses both positive and negative SAM periods. Therefore, changes in atmospheric transport may not be the primary cause of the increased biomass burning in Talos Dome.

(iii) The Patagonian fire season occurs during the austral spring and summer where major fire years coincide with severe droughts in December (Mundo et al. 2017). Arienzo et al. (2017) evaluated the difference in insolation during the growing (February) and burning season (October) at 15°S, with a positive correlation with the black carbon (BC) record in Antarctica, during the entire Holocene. BC is produced by both fossil fuel and biomass burning, where the dominant pre-industrial source is fire activity. Therefore, insolation may influence both the burning and growing seasons. Fire occurrence in Patagonia is influenced prevalently by the growing season in the northern regions and prevalently by the summer droughts in the southern (Mundo et al., 2017). As different drivers affect fires in Patagonia, we opted to consider the insolation during the period between October and February to take into account to the overall contributions. A significant positive correlation with solar radiation in October-February at 45°S (from Laskar et al. 2004) (r (95% CI) =0.378(0.280, 0.494); see also Fig.5) is depicted by a long-term increase up to approximately 2500 yr BP. The cooling period centered at about 2000 yr BP coincides with maximum values of October-February solar radiation and a shift to drier conditions in South America, characterized by initial conditions with available fuel, thus enhancing biomass burning in Patagonia in this particular time period.

Although the levoglucosan record is consistent with October-February insolation at 45°S, our levoglucosan results differ from the insolation at 15°S used by Arienzo et al. (2017) to depict the growing and burning seasons in South American regions potentially influencing fire histories in Antarctic ice cores. This difference may be due to the atmospheric lifetimes and/or deposition of the two proxies. As previously mentioned, studies of the atmospheric lifetime of levoglucosan range between a few days (Hoffman et al., 2010; Hennigan et al., 2010) to a few weeks (Bai et al., 2013; Slade and Knopf, 2013), and levoglucosan originating at 15°S may not reach Antarctica. However, combining ice core BC with multiple climate model simulations result in BC lifetimes ranging from 3.9 to 15.2 days (Lee et al., 2012), with 4 days as a reasonable BC lifetime under present atmospheric conditions (Bauer et al., 2013), suggesting that 15°S is also a distant source for BC. As BC is primarily deposited through wet deposition (Flanner et al., 2007) changes in hydroclimate primarily affect BC deposition in Antarctica (Areinzo et al., 2017). However, as levoglucosan is deposited by both wet and

dry deposition, changes in the hydroclimate are not the only possible influence on levoglucosan flux (Bertler et al., 2018; Zennaro et al., 2015).

Arienzo et al. (2017) argue that BC in Antarctica reflects Southern Hemisphere biomass burning rather than atmospheric transport as BC is well mixed in the Antarctic atmosphere. BC concentration and flux are influenced by the amount of precipitation as BC is deposited through wet deposition (Flanner et al. 2007). A continuous record of BC and ammonium from the coastal east Antarctic B40 ice core (see Fig.1) demonstrates a peak in biomass burning centered around ~2000 yr BP, which is consistent with our results. One of the strengths of BC is that it can be continuously determined at high resolution in unbroken ice. However, the brittle ice section of the west Antarctic WAIS Divide ice core results in a semi-continuous record of BC and ammonium from ~6000 yr BP to the present (Arienzo et al., 2017)., hindering a comparison with our record. Arienzo et al.'s (2017) argument that the biomass burning record in Antarctica primarily reflects changes in biomass burning rather than in atmospheric transport is consistent with the interpretation of the levoglucosan record in Talos Dome (see point (ii)).

### 3.5 Possible anthropogenic influence?

This Talos Dome levoglucosan record is limited by the available ice core samples and does not extend to the present, thereby not covering the time period of major regional population shifts, yet covering the time period of the advent of agriculture on a global scale. Several studies examine the anthropogenic influence on global to regional fire activity due to agriculture in the Mid-Late Holocene (e.g. Broeker and Stocker,2006; Joos et al., 2004; Singarayer et al., 2011; Mitchell et al., 2013, Kaplan et al., 2009 and 2011). Although the first agricultural practices in South America likely appeared ~10,000 yr BP, the development of agriculture in the Southern Hemisphere was limited with respect to the Northern Hemisphere (Martin and Sauerborn, 2013). The Hyde 3.2 database demonstrates that in the possible fire source regions of Patagonia and Southern Africa the demographic density was between 1-5 inhabitant per $km^2$ and even less than 1 inhabitants/$km^2$ in Australia until ~ 1000 yr BP (Goldewijk, 2016). Human populations in the region fluctuated wildly from 1000 yr BP to present due to European settlement and the associated decimation of indigenous populations. The human impact on fire activity is not directly proportional to population density, where small bands of humans can substantially change their environment, such as in the case of rapid burning of New Zealand forests (McWethy et al., 2010 and 2014). However, the Talos Dome ice core record from 6000 yr BP to ~750 yr BP currently does not provide clear evidence that the fire record may be strongly affected by anthropogenic activities during the mid-late Holocene, although we cannot exclude at least a partial influence.

### 4 Conclusion

We determined the specific biomass burning biomarker levoglucosan and potassium in ice core from the TALDICE during the mid-late Holocene (750-6000 yr BP). The comparison between levoglucosan and the multi-source potassium suggested that potassium can increase the information obtained from a

single fire proxy, although $K_{bb}$ is often best used in conjunction with other biomass burning markers. The levoglucosan record is characterized by a long-term increase with higher rates starting at~4000 yr BP and higher peaks between 1500 and 2500 yr BP. These peaks are consistent with other biomass burning records from Antarctica (Arienzo et al., 2017). Comparisons with regional charcoal syntheses suggested a primary contribution from southern South American fires rather than Australian biomass burning.

The location of Talos Dome provides an ideal situation to compare the biomass burning records of this ice core with ice cores from the interior of the continent as well as on the Antarctic Peninsula. Air masses reaching Antarctica are influenced by ENSO and SAM, where the relative importance of these atmospheric phenomenon change through time and space, and therefore may influence the distribution of fire aerosols reaching each ice core location. Talos Dome is one end member of the Ross Sea Dipole (Bertler et al., 2018) where during the last thousand years, the stable isotopes of precipitation in TALDICE are in sync with $SAM_A$. If this connection between $SAM_A$ and TALDICE stable isotopes remains consistent throughout the mid-late Holocene, then our results demonstrate that biomass burning aerosols reaching TALDICE do not depend on the most prominent atmospheric phenomenon affecting the site. This coastal location and documented dipole (Bertler et al., 2018) suggests that the air masses affecting TALDICE are not always the same as those affecting internal or peninsular Antarctic sites. Changes in biomass burning, rather than changes in atmospheric transport are the primary influence on fire aerosols recorded in TALDICE. We determine that a generalized cooling period in the Southern Hemisphere ~2000 yr BP coincides with a shift to drier conditions in South America, characterized by initial conditions with available fuel, that, in turn, created favourable conditions for fire. Finally, although we cannot exclude a possible anthropogenic influence on the SH fire history, we did not find clear evidence for supporting this occurrence.

**Acknowledgments**

This work was financially supported by the Early Human Impact ERC Advanced Grant of the European Commission's VII Framework Program, grant number 267696, by Programma Nazionale di Ricerche in Antartide (PNRA) Grant: PEA 2013/B2.05, and by European Union Marie Curie IIF Fellowship (MIF1-CT-2006-039529, TDICOSO) within the VII Framework Program. The Talos Dome Ice Core Project (TALDICE), a joint European programme, is funded by national contributions from Italy, France, Germany, Switzerland and the United Kingdom. Primary logistical support was provided by PNRA at Talos Dome. The authors gratefully acknowledge the help of ELGA LabWater in providing the PURELAB Pulse and PURELAB Flex that produced the ultrapure water used in these experiments. The authors thank Barbara Delmonte for providing raw Talos Dome dust data. Any use of trade, firm, or product names is for descriptive purposes only and does not imply endorsement by the U.S. Government.

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

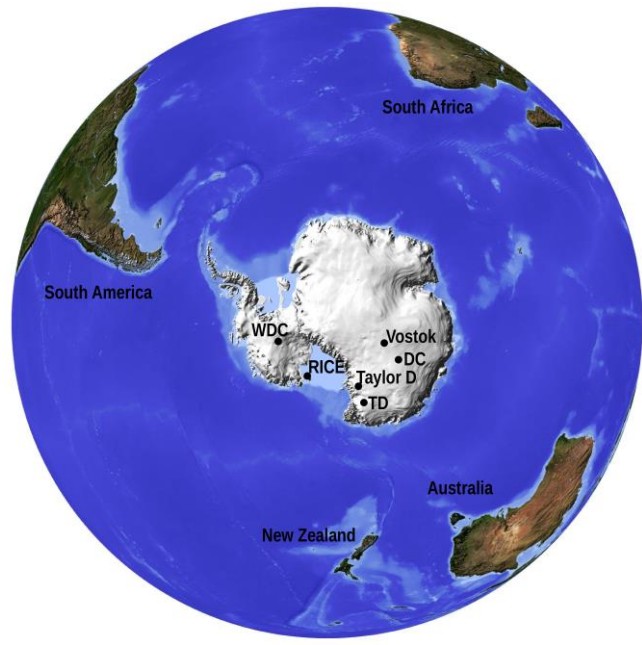

**Figure 1: Map of the Antarctic region including the Talos Dome, Vostok, Taylor Dome, Dome C, RICE and WAIS Divide ice core sites mentioned in this paper.**

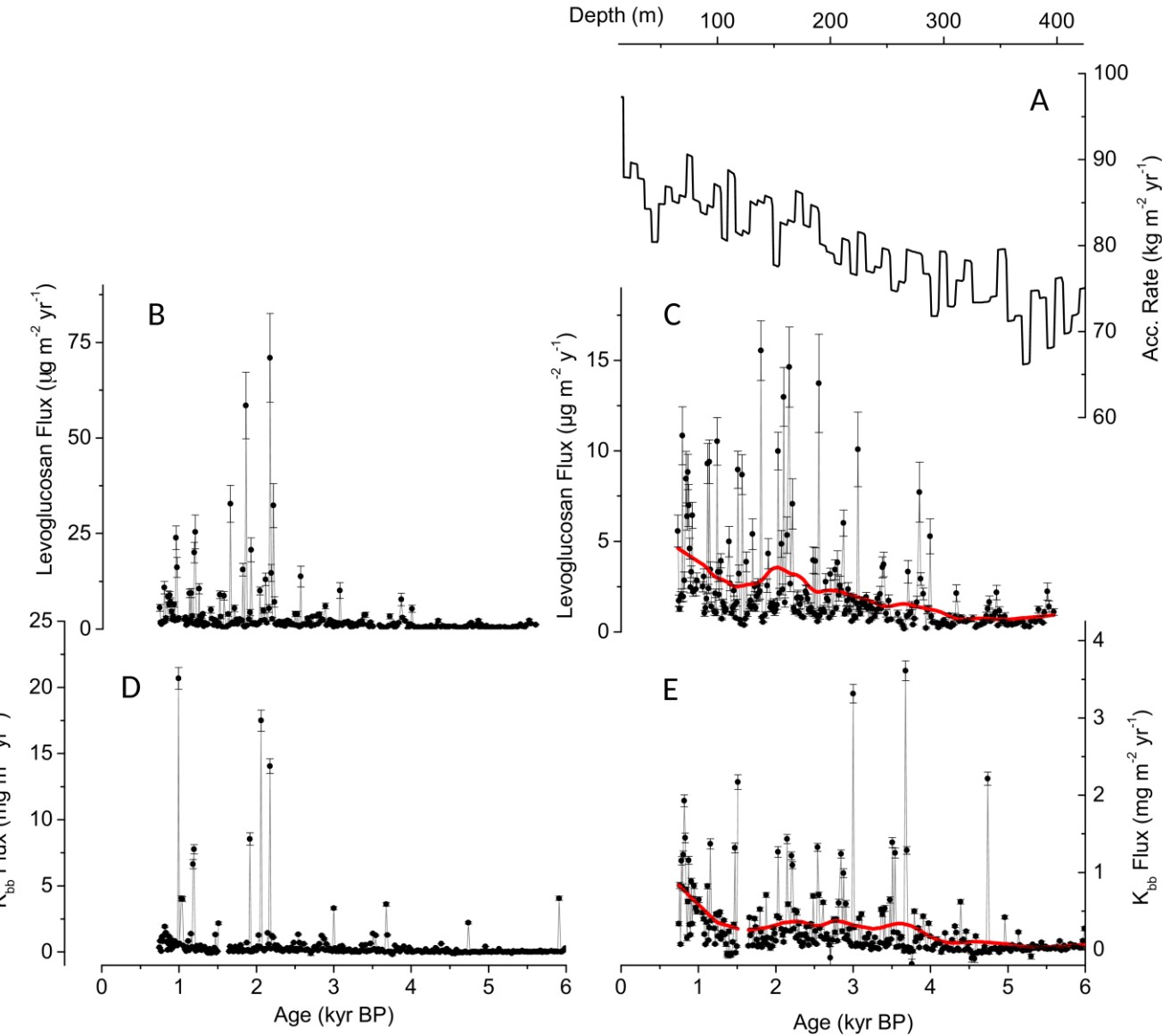

Figure 2: (A) Accumulation rate (from Veres et al. 2013). Complete (B) and without anomalies (C) levoglucosan record (LOWESS smoothing with SPAN parameter 0.2 (red)). Complete (D) and without anomalies (E) $K_{bb}$ record (LOWESS smoothing with SPAN parameter 0.2 (red)) from TALDICE

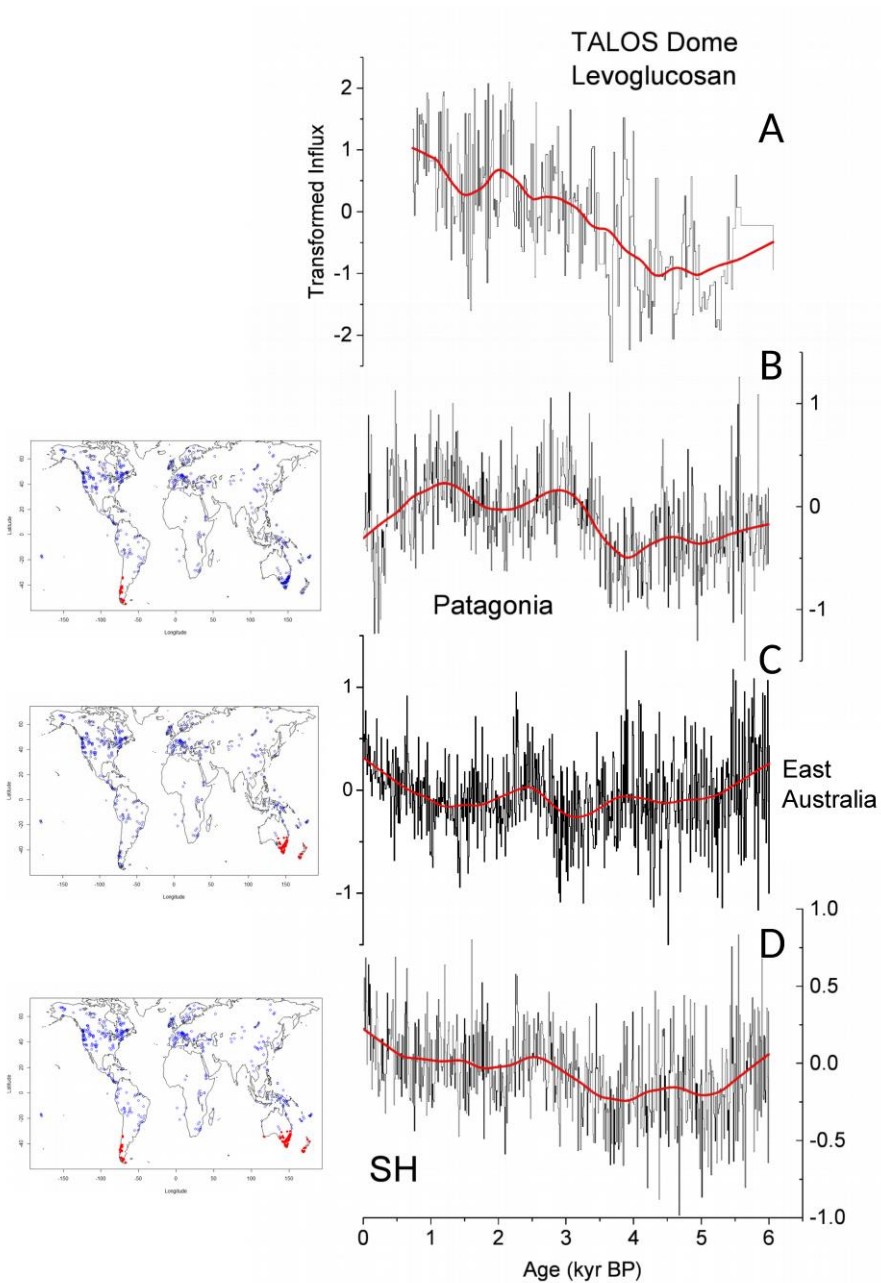

**Figure 3: Box-Cox transformed levoglucosan influx and charcoal synthesis (bins= 20 years) from Patagonia, East Australia and Southern Hemisphere synthesis from GCD. LOWESS smoothing (in red) with SPAN parameters 0.2.**

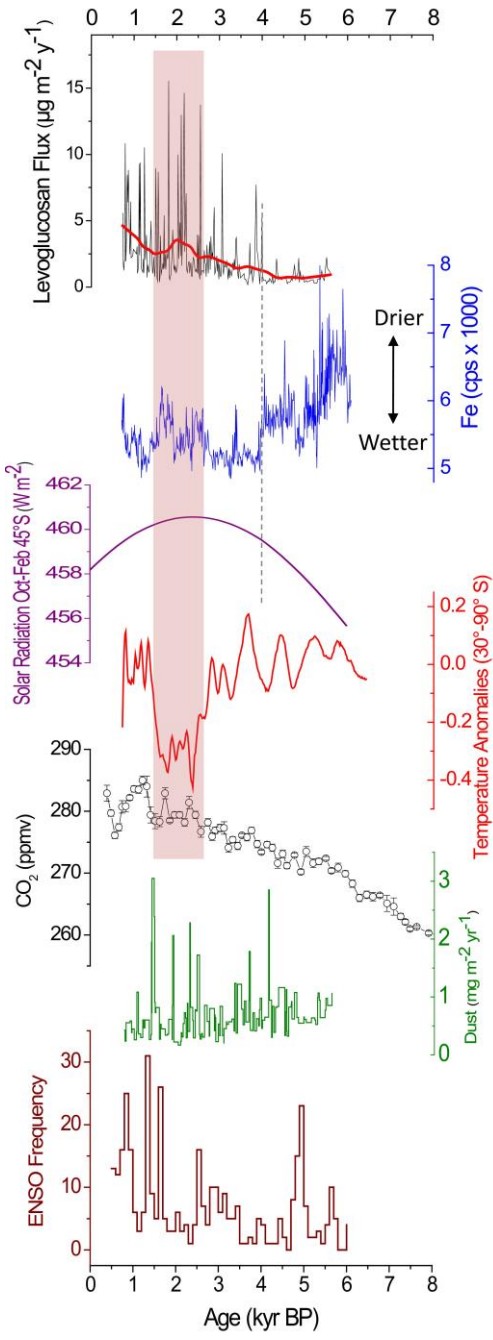

**Figure 4: Levoglucosan flux, Fe content in marine sediments from SouthernChile (from Lamy et al. 2001); October-February 45°S solar radiation (from Laskar et al. 2004), Temperature anomalies between 30° and 90° S (Marcott et al. 2013), $CO_2$ from Taylor Dome (Indermuhle et al. 1999), Dust from Talos Dome (Albani et al. 2002) and ENSO 100 yr frequency (Moy et al. 2002).**

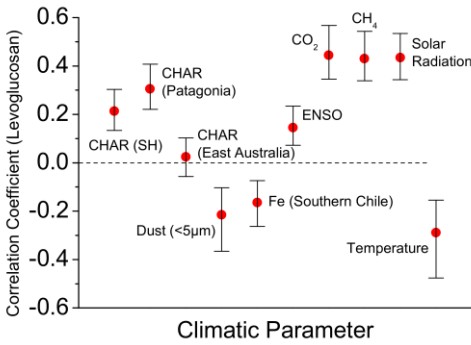

**Figure 5: Correlation between levoglucosan recorded at Talos Dome (log transformed) and charcoal synthesis (CHAR) from Patagonia, East Australia and Southern Hemisphere, dust particles at Talos Dome (from Albani et al. 2012), Iron record from Southern Chile (Lamy et al. 2001), Carbon dioxide from Taylor Dome (Indermuhle et al. 1999), ENSO 100 yr frequency (Moy et al. 2002), 30°S December solar radiation (from Berger and Loutre 1991). (95% confidence interval (CI)).**

**Table 1: Levoglucosan and $K_{bb}$ peak anomalies. Levoglucosan and $K_{bb}$ values are minmax scaled. n.a. = not available ; n.s. = these values were not anomalies.**

| Group | TD Sample | Age yr BP | Levoglucosan | $K_{bb}$ | Match |
|---|---|---|---|---|---|
| | 108 | 945 | 0.34 | **n.s.** | No |
| | 109 | 957 | 0.23 | **n.a** | |
| | 112 | 994 | **n.a** | 1.00 | |
| I | 115 | 1031 | **n.a** | 0.21 | |
| | 116 | 1043 | **n.a** | 0.20 | |
| | 127 | 1180 | 0.28 | 0.33 | **Yes** |
| | 128 | 1193 | 0.36 | 0.39 | **Yes** |
| II | 163 | 1647 | 0.46 | **n.s.** | No |
| | 178 | 1849 | 0.82 | **n.s.** | No |
| III-a | 183 | 1916 | 0.29 | 0.39 | **Yes** |
| | 193 | 2060 | **n.a** | 0.79 | |
| | 200 | 2160 | 1.00 | **n.s.** | No |
| III-b | 201 | 2174 | **n.s.** | 0.63 | No |
| | 203 | 2202 | 0.46 | **n.s.** | No |
| | 299 | 5911 | **n.a** | 0.15 | |