# Peer review of "High latitude Southern Hemisphere fire history during the mid-late Holocene (750-6000 yr BP)"

_Climate of the Past, 2017_

## Referee Comment (RC1) · Anonymous Referee #1 · 5 Feb 2018

This manuscript reports on levoglucosan and the calculated fine (i.e., non sea-salt and non dust) potassium fraction measured on sub-samples from a Talos Dome (Antarctica) ice core spanning the second part of the Holocene. The aim of the work is to reconstruct past fire activity in the southern hemisphere.

The data presented in the manuscript possibly contain information in view to better understand past variability of fire activity. This topic is clearly relevant for Climate of the Past journal. As it stands the manuscript, however, requires major revisions on several key aspects and a re-evaluation prior to publication.

First, the wording "We reconstructed the high latitude Southern Hemisphere fire history by using the specific biomarker levoglucosan and potassium in ice cores from the TALDICE during the Mid-Late Holocene (750-6000 yr BP)» is clearly an overstatement since your data are discontinuous (only 15% of the ice core was measured). Please, make this point more clear in the manuscript and change the wording in your discussion.

Second, I would like to see in the manuscript the raw data (concentrations and not only calculated deposition flux) for levoglucosan and potassium. If my estimate is correct, you calculated fine potassium concentrations in the order of 0.2 ppb. I strongly recommend to the authors to show concentrations of potassium, sodium, and iron and error propagation estimates. Only with that the reviewers (and the readers) can evaluate to robustness of the potassium approach. Also concerning the discrepancies between levoglucosan and potassium: line 8 page 7: what other sources for potassium in the pre-industrial atmosphere (apart from sea-salt and dust) you have in mind here ????

Third, you have ignored the study of black carbon from Antarctic ice covering the Holocene from Arienzo et al. (2017) published in the JGR in 2017. Even though the ice core discussed in this recent paper is from another Antarctic site, it is needed to refer to this previous work and discuss similarity and difference with your work. Note that this JGR paper is based on a continuous back carbon record. I attached below the abstract, showing how deep was the discussion of this record in terms of climatic condition changes in the south hemisphere and particularly in South America.

"Black carbon (BC) and other biomass-burning (BB) aerosols are critical components of climate forcing, but quantification, predictive climate modeling, and policy decisions have been hampered by limited understanding of the climate drivers of BB and by the lack of long-term records. Prior modeling studies suggested that increased Northern Hemisphere anthropogenic BC emissions increased recent temperatures and regional precipitation, including a northward shift in the Intertropical Convergence Zone (ITCZ). Two Antarctic

ice cores were analyzed for BC, and the longest record shows that the highest BC deposition during the Holocene occurred ~8–6 k years before present in a period of relatively high austral burning season and low growing season insolation. Atmospheric transport modeling suggests South America (SA) as the dominant source of modern Antarctic BC and, consistent with the ice core record, climate model experiments using mid-Holocene and preindustrial insolation simulate comparable increases in carbon loss due to fires in SA during the mid-Holocene. SA climate proxies document a northward shifted ITCZ and weakened SA Summer Monsoon (SASM) during this period, with associated impacts on hydroclimate and burning. A second Antarctic ice core spanning the last 2.5 k years documents similar linkages between hydroclimate and BC, with the lowest deposition during the Little Ice Age characterized by a southerly shifted ITCZ and strengthened SASM. These new results indicate that insolation-driven changes in SA hydroclimate and BB, likely linked to the position of the ITCZ, modulated Antarctic BC deposition during most of the Holocene and suggests connections and feedbacks between future BC emissions and hydroclimate."

In conclusion, I would like to review a more adequate version of this manuscript in which authors report concentrations, provide error propagation estimates for fine potassium, and discuss in depth their finding with the continuous black carbon record from Arienzo et al. 2017.

Arienzo, M. M., J. R. McConnell, L. N. Murphy, N. Chellman, S. Das, S. Kipfstuhl, and R. Mulvaney (2017), Holocene black carbon in Antarctica paralleled Southern Hemisphere climate, J. Geophys. Res. Atmos., 122, 6713–6728, doi:10.1002/2017JD026599.

End of the review

---

## Referee Comment (RC2) · Anonymous Referee #2 · 9 Feb 2018

This manuscript describes the first long-term record of levoglucosan from Antarctica. Levoglucosan, a specific biomarker of biomass burning, was measured from 750-6000 yr BP. The levoglucosan record was compared to Kbb of the same ice core. The authors conclude that the Kbb record supports the Levoglucosan record and for the interpretation rely on the Levoglucosan record. The authors compare the levoglucosan record to charcoal records and demonstrate the greatest correlation to Patagonia charcoal. The authors propose that in general, the levoglucosan record demonstrate a linkage between climatic changes in Patagonia and biomass burning. Lastly the authors focus on a peak in Levoglucosan centered at ∼2,000 yr BP.

In general, I found this manuscript interesting to read and the manuscript is well written. I appreciate that this the first long-term ice core record of levoglucosan from Antarctica.

[Figure]

I recommend the authors do not focus the manuscript on the individual spikes observed in their records as these spikes have not be replicated in other Antarctic ice cores and therefore the spikes may be driven by degradation/transport rather than changes at the source. I think the manuscript would benefit from removing repetitive sections in Section 3 while adding additional discussion in other parts of the manuscript. In addition, this manuscript would benefit from a clearer explanation of the climatic mechanisms driving the observed correlations.

Specific comments: Introduction: It is worth noting on page three that while few Levoglucosan records exist in Antarctica, several studies on the monitoring of biomass burning byproducts (Hu et al., 2013 Scientific Reports; Pereira et al., 2006; Weller et al., 2013; Wolff and Cachier, 1998; Fiebig et al., 2009; Hara et al., 2010) have been conducted in Antarctica. In addition, other biomass burning proxy records have been published from Antarctic ice cores (ex. Pasteris et al., 2014; Bisiaux et al., 2012; Arienzo et al., 2017) and are also worth mentioning here.

Section 2.1: How was LoQ determined?

Section 2.2: What are the errors on the ages at this depth in the ice core? Since Kbb is introduced in this section, I would suggest moving equation 1 and the explanation here.

Figure 2: I found figure 2 very confusing in particular since the X-axis and Y-axis vary per plot. I would recommend plotting the fluxes on the same X axis and possibly changing the color of the lines to indicate the different data being plotted. I would also include accumulation rate in this plot to demonstrate that the levoglucosan and Kbb flux records are not dominated by variations in accumulation. Lastly, in both A and B a LOWES smoothing is shown with two varying SPAN parameters. I don't understand why the red curve in panel A is lower frequency and the red curve in panel B is higher frequency (when compared to the black cure in each panel). I also noticed that in figure 3, a different SPAN is used and then in figure 4 a 250 year average window is shown.

I would recommend choosing one smoothing parameter and showing the results using that smoothing in all figures. This allows the reader to easily compare between the various plots.

Section 3.1: In this section, the authors demonstrate at the end of page 5 to the second paragraph on Page 6 that there are significant uncertainties associated with interpreting the spikes observed in the Levoglucosan record. This provides the justification for removing the large spikes when discussing the overall trend of the Levoglucosan record. However, later in this section and in other parts of the manuscript the authors discuss the spikes as potential evidence of specific fires. I would caution the authors about interpreting the spikes observed in the record as changes in the source area, considering that there are very few that agree between the levoglucosan and Kbb records and that transport is potentially a large driver of the spikes.

Section 3.3: I suggest renaming this section "Biomass Burning Sources" I appreciate that the authors discuss the potential for other biomass burning sources in this section (lines 19-29, p. 9). However, I would encourage the authors to preface this with a discussion of the comparison between the Patagonia and the TALOS record as several differences and similarities exist. For example at 4.5 kyr the TALOS Levoglucosan begins to increase, while the Patagonia record is decreasing at this time. This could then be followed by a discussion of other possible biomass burning sources. Lines 26 to 29 p. 9 would be more appropriate in section 3.4. Section 3.3 addresses the potential sources of biomass burning, not the impact of climate on the charcoal records. Lines 19-26 are relevant as they explain the heterogeneity of fire in Australia so maybe something like "Similar heterogeneities are not observed in Patagonia (Ref)" would be more appropriate instead.

Section 3.4: I would rename this section "Drivers of biomass burning" This section overlaps with section 3.6. I would consider merging the two sections to make one section that very clearly states what the proposed drivers of biomass burning and the mechanisms driving these changes are. Also reorganizing would allow the authors to

discuss figure 4 in its entirety in one section.

Section 3.5: Given the uncertainties in the sources of levoglucosan and transport variations I would be cautious about attributing the spikes in the Levoglucosan record to specific fire events within charcoal records.

Section 3.6: As stated above, the overall discussion of climate/fire linkages would be much stronger if this section was merged with section 3.4. For example, P.11 line 14 the authors state that "variable and generally wet conditions prevailed in South America after 4000 yr BP" which would very nicely flow with the discussion in section 3.4. While this section includes a very nice statistical comparison to various climatic indicators, the mechanistic linkages between the climate and biomass burning (and hence Levoglucosan) are unclear. For example, what climatic mechanism would link 30 degree solar radiation in December to Patagonia biomass burning/Levoglucosan? Additional discussion with references would strengthen this section of the article. In the discussion about ENSO, is there evidence in the modern for an impact of ENSO on burning in Patagonia? It might be worth noting, other proxy biomass burning records from Antarctica (ex Bisiaux et al., 2012, ACP) demonstrate a modern relationship to ENSO. In the discussion of the drivers of the peak Levoglucosan (P. 12 lines 1-7), I appreciate the discussion here.

Conclusions: The statement "Potassium was analyzed in order to provide a more complete biomass burning record through the comparison with another fire proxy" should be reworded. I would suggest instead "The comparison between levoglucosan and the multi-sourced Kbb suggests that Kbb is best used in conjunction with other biomass burning markers."

Technical/minor comments: Page 5, line 16: This would be appropriate in the background.

While changes in the westerly wind belt would have changed climate in Patagonia, is it possible that changes in the regional scale winds would also have impacted transport

of Levoglucosan to Antarctica?

Why is the levoglucosan record compared to Southern Hemisphere charcoal (page 10 line 27-28) when in section 3.3 the source was determined to be most likely Patagonia?

P. 11 lines 10-12: this sentence is unclear.

P.11 line 16: "This 1000-year peak in levoglucosan. . ." I would encourage the authors to define the age range of the peak referenced here.

A reference is needed to support the statement "the cooling period coincides with a shift to drier conditions in South America (page 12 line 6-7)".

Figure 2: The Y axis for plot 2B does not have a complete label. The caption in missing a parentheses.

Figure 4: This plot would be clearer if the y-axes were labeled rather than having the labels floating within the plot. Does the 250 yr average window include the spikes? This is unclear since the spikes are plotted.

---

## Editor Comment (EC1) · EW Wolff (Editor) · 16 Feb 2018

The discussion period for this paper is nearly complete. You have two substantial reviews. When the discussion period is over, please post author comments responding to each comment made by the two reviewers and indicate how you would change the manuscript to address them. I will then be asked to give an editorial decision to let you know whether you are encouraged to submit a revised version. Although reviewer one is brief, they do ask for rather substantial changes before they feel they can evaluate the paper. So please do consider this - most likely the paper will need to be peer-reviewed again when you submit a new version.

---

## Author Comment (AC1) · 22 Mar 2018

Thank you for the opportunity to respond to the suggestions from both of the reviewers. We have substantially changed the paper, including combining sections in the discussion, expanding out discussion of the possible atmospheric influences, and comparing our results with other Antarctic biomass burning records. We also substantially altered Figure 4, accordingly (attached file). Due to these changes, we rewrote the abstract and conclusions. We can understand if these changes require the paper to be sent out for review again, and welcome any opportunity to improve our work.

[Figure]

$m^{-2} y^{-1})$

0

15

---

## Author Comment (AC2) · 22 Mar 2018

This manuscript describes the first long-term record of levoglucosan from Antarctica. Levoglucosan, a specific biomarker of biomass burning, was measured from 750-6000 yr BP. The levoglucosan record was compared to Kbb of the same ice core. The authors conclude that the Kbb record supports the Levoglucosan record and for the interpretation rely on the Levoglucosan record. The authors compare the levoglucosan record to charcoal records and demonstrate the greatest correlation to Patagonia charcoal. The authors propose that in general, the levoglucosan record demonstrate a linkage between climatic changes in Patagonia and biomass burning. Lastly the authors focus on

a peak in Levoglucosan centered at âĹij2,000 yr BP. In general, I found this manuscript interesting to read and the manuscript is well written. I appreciate that this the first long-term ice core record of levoglucosan from Antarctica.

I recommend the authors do not focus the manuscript on the individual spikes observed in their records as these spikes have not be replicated in other Antarctic ice cores and therefore the spikes may be driven by degradation/transport rather than changes at the source. I think the manuscript would benefit from removing repetitive sections in Section 3 while adding additional discussion in other parts of the manuscript. In addition, this manuscript would benefit from a clearer explanation of the climatic mechanisms driving the observed correlations.

Specific comments:

Introduction: It is worth noting on page three that while few Levoglucosan records exist in Antarctica, several studies on the monitoring of biomass burning byproducts (Hu et al., 2013 Scientific Reports; Pereira et al., 2006; Weller et al., 2013; Wolff and Cachier, 1998; Fiebig et al., 2009; Hara et al., 2010) have been conducted in Antarctica. In addition, other biomass burning proxy records have been published from Antarctic ice cores (ex. Pasteris et al., 2014; Bisiaux et al., 2012; Arienzo et al., 2017) and are also worth mentioning here.

Reply:Thank you for this suggestion. We included these references in the introduction section in the following sentences:

"However, Southern Hemisphere levoglucosan records still leave much to be explored. Antarctic levoglucosan records will likely differ from Arctic records due to the substantially smaller land masses surrounding near the ice sheet and the long distances required for atmospheric transport of biomass burning material. The southernmost tip of Patagonia, the closest continental landmass to Antarctica, only extends to ∼55°S, whereas the Arctic contains the largest land masses in the world. Although long-term levoglucosan records have never been reported for Antarctic ice cores, several studies determine other biomass burning by-products (i.e. secondary organic aerosol and black carbon) during the last few decades and centuries (Wolff and Cachier, 1998, Fiebig et al. 2009, Hara et al. 2010, Hu et al. 2013, Weller et al. 2013, Pasteris et al. 2014) as well as during the Holocene (Arienzo et al. 2017)."

Section 2.1: How was LoQ determined?

Reply: We added the following discussion to page 4 to better explain how we determined the LOQ:

"The instrumental limit of quantification (LoQ) for levoglucosan was 4 pg mL-1, determined following the analytical method reported in Gambaro et al. 2008. From an analytical point of view, a reliable procedural LoQ is difficult to determine in this case, due to the lack of a suitable aqueous matrix. The glacier water often contains lower concentrations of levoglucosan than the ultra-pure laboratory water, thereby complicating obtaining a true LoQ. However, this method only requires as a few pre-analytical procedures, where these steps are always performed in a dedicated clean room, we therefore use the instrumental values as the LoQ."

Section 2.2: What are the errors on the ages at this depth in the ice core? Since Kbb is introduced in this section, I would suggest moving equation 1 and the explanation here.

Reply:We now include a supplementary file with all of the raw data and associated age errors (rawdata.xls). We followed your suggestion to move equation 1 to section 2.2.

Figure 2: I found figure 2 very confusing in particular since the X-axis and Y-axis vary per plot. I would recommend plotting the fluxes on the same X axis and possibly changing the color of the lines to indicate the different data being plotted. I would also include accumulation rate in this plot to demonstrate that the levoglucosan and Kbb flux records are not dominated by variations in accumulation. Lastly, in both A and B a LOWES smoothing is shown with two varying SPAN parameters. I don't understand

why the red curve in panel A is lower frequency and the red curve in panel B is higher frequency (when compared to the black cure in each panel). I also noticed that in figure 3, a different SPAN is used and then in figure 4 a 250-year average window is shown. I would recommend choosing one smoothing parameter and showing the results using that smoothing in all figures. This allows the reader to easily compare between the various plots.

Reply: We completely revised Figure 2, following your suggestions. We now use a SPAN parameter of 0.2 in each figure. We also include a graph of the accumulation in Figure 2.

Section 3.1: In this section, the authors demonstrate at the end of page 5 to the second paragraph on Page 6 that there are significant uncertainties associated with interpreting the spikes observed in the Levoglucosan record. This provides the justification for removing the large spikes when discussing the overall trend of the Levoglucosan record. However, later in this section and in other parts of the manuscript the authors discuss the spikes as potential evidence of specific fires. I would caution the authors about interpreting the spikes observed in the record as changes in the source area, considering that there are very few that agree between the levoglucosan and Kbb records and that transport is potentially a large driver of the spikes.

Reply: We agree with the fact that spikes are not very informative by themselves and that they may lead to incorrect assumptions. In the original version, we tried to emphasize the fact that spikes must be considered carefully. In order to deemphasize the spikes, we reduced the discussion of spikes, and moved this discussion to the end of Section 3.1. This paragraph is now the following:

"The attribution of the levoglucosan spikes to individual large fire events is difficult to assess as atmospheric transport and stability may alter the signal, resulting in an amplification of modest fires. A similar situation occurs for individual peaks in charcoal records as these spikes can either be intense local fire events or can result from an

increase in short-term transport. For example, the high charcoal signals observed in Laguna Padre Laguna (Argentina) between ∼1500 and ∼2000 yr BP and at Laguna Zeta (Argentina) between ∼2000 and ∼2500 yr BP (Iglesias and Whitlock, 2014) are consistent with the spikes observed in the levoglucosan record, as well as intense fire episodes recorded between ∼2100 and ∼2300 yr BP in the Wingecarribee Swamp (Southern Australia (de Montford, (2008); ID site=857 in the GCD) and at 2160 yr BP in Eweburn Bog (New Zealand, (Ogden et al. 1998)); ID site=441 in the GCD). A possible correspondence with levoglucosan and these individual charcoal spikes is only speculative. Comparing trends in long-term fire activity, as identified by smoothed levoglucosan records and charcoal syntheses, is more indicative of changes in biomass burning as opposed to comparing individual, likely localized, events. We therefore only discuss biomass burning trends rather than individual spikes from this point forward."

Section 3.3: I suggest renaming this section "Biomass Burning Sources" I appreciate that the authors discuss the potential for other biomass burning sources in this section (lines 19-29, p. 9). However, I would encourage the authors to preface this with a discussion of the comparison between the Patagonia and the TALOS record as several differences and similarities exist. For example, at 4.5 kyr the TALOS Levoglucosan begins to increase, while the Patagonia record is decreasing at this time. This could then be followed by a discussion of other possible biomass burning sources. Lines 26 to 29 p. 9 would be more appropriate in section 3.4.

Reply: We changed the title of this section to "Biomass Burning Sources" and expanded the discussion of the similarities and differences in the records into the following paragraph. In addition, we combined Sections 3.2 and 3.3 (please see the following point) to better continue the discussion of possible fire sources.

"Talos Dome Levoglucosan and southern South America charcoal records (Fig.3) both depict an increasing long-term trend in biomass burning from 4000 to 750 yr BP, although these trends do not perfectly coincide. These records differ both before and after these ∼3000 years of similarity. The Talos Dome record begins its major and sustained increase in fire activity around 4500 yr BP, while the southern South American record is decreasing during this time period. The southern South America and East Australian records substantially diverge after 750 yr BP, where the southern South American fires decrease and the East Australian and SH fires both markedly increase. The Talos Dome levoglucosan record does not extend to the present, and so comparisons with this major difference in regional fire sources is not possible."

Section 3.3 addresses the potential sources of biomass burning, not the impact of climate on the charcoal records. Lines 19-26 are relevant as they explain the heterogeneity of fire in Australia so maybe something like "Similar heterogeneities are not observed in Patagonia (Ref)" would be more appropriate instead.

Reply: We now concentrate the only the sources of biomass burning in this section, and have moved what was Section 3.3 to the end of Section 3.2. This paragraph is now the following:

"These results may suggest a higher contribution from southern South American fires rather than Australian biomass burning over centennial to millennial timescales between 4000 to 750 yr BP, although the contribution from East Australia cannot be completely neglected. However, it is also possible that this observation may be due to the higher regional variability in East Australian fire activity (Mooney et al., 2011). During the Mid-Holocene, sites in the far south of South-Eastern Australia showed higher fire activity due to a shift toward more moisture-stressed vegetation, while sites in the Southern Tablelands and east of the Great Dividing Range showed less fire activity with the region supporting relatively moisture-demanding vegetation (Pickett et al., 2004; Mooney et al., 2011). Thus, over a wider spatial scale, these contributions tend to cancel one another. As a result, although the South-Eastern Australia charcoal synthesis is characterized by higher values at about 6000 yr BP, during the Mid-Holocene, only small local increases at $\sim$4000 y BP and $\sim$2500 yr BP were observed. The southern South American fire synthesis is generally more homogenous, although some regional spatial variability between northern and southern Patagonia does exist (Mundo et al.

2017). The southern South American region defined in these charcoal syntheses extends Patagonia to the Argentinian Pampas, encompassing grassland, steppe, desert and forest vegetation."

Section 3.4: I would rename this section "Drivers of biomass burning" This section overlaps with section 3.6. I would consider merging the two sections to make one section that very clearly states what the proposed drivers of biomass burning and the mechanisms driving these changes are. Also reorganizing would allow the authors to discuss figure 4 in its entirety in one section.

Reply: We merged sections 3.4 and 3.6, and omitted section 3.5 in order to improve the flow of the paper.

Section 3.5: Given the uncertainties in the sources of levoglucosan and transport variations I would be cautious about attributing the spikes in the Levoglucosan record to specific fire events within charcoal records.

Reply: We omitted section 3.5, but did keep the following short discussion on individual spikes, which is now at the end of section 3.1. We kept the following paragraph in order to primarily address the fact that we will only examine long-term trends in the data:

"The attribution of the levoglucosan spikes to individual large fire events is difficult to assess as atmospheric transport and stability may alter the signal, resulting in an amplification of modest fires. A similar situation occurs for individual peaks in charcoal records as these spikes can either be intense local fire events or can result from an increase in short-term transport. For example, the high charcoal signals observed in Laguna Padre Laguna (Argentina) between ∼1500 and ∼2000 yr BP and at Laguna Zeta (Argentina) between ∼2000 and ∼2500 yr BP (Iglesias and Whitlock, 2014) are consistent with the spikes observed in the levoglucosan record, as well as intense fire episodes recorded between ∼2100 and ∼2300 yr BP in the Wingecarribee Swamp (Southern Australia (de Montford, (2008); ID site=857 in the GCD) and at 2160 yr BP in Eweburn Bog (New Zealand, (Ogden et al. 1998)); ID site=441 in the GCD). A possible correspondence with levoglucosan and these individual charcoal spikes is only speculative. Comparing trends in long-term fire activity, as identified by smoothed levoglucosan records and charcoal syntheses, is more indicative of changes in biomass burning as opposed to comparing individual, likely localized, events. We therefore only discuss biomass burning trends rather than individual spikes from this point forward."

Section 3.6: As stated above, the overall discussion of climate/fire linkages would be much stronger if this section was merged with section 3.4. For example, P.11 line 14 the authors state that "variable and generally wet conditions prevailed in South America after 4000 yr BP" which would very nicely flow with the discussion in section 3.4. While this section includes a very nice statistical comparison to various climatic indicators, the mechanistic linkages between the climate and biomass burning (and hence Levoglucosan) are unclear. For example, what climatic mechanism would link 30 degree solar radiation in December to Patagonia biomass burning/Levoglucosan? Additional discussion with references would strengthen this section of the article. In the discussion about ENSO, is there evidence in the modern for an impact of ENSO on burning in Patagonia? It might be worth noting, other proxy biomass burning records from Antarctica (ex Bisiaux et al., 2012, ACP) demonstrate a modern relationship to ENSO. In the discussion of the drivers of the peak Levoglucosan (P. 12 lines 1-7), I appreciate the discussion here.

Reply: We agree that merging sections 3.4 and 3.6 help with the flow of the paper. As stated in one of the previous replies, we merged these sections and omitted section 3.5. We also agree that using insolation for 30°S December was not the best choice and now use solar radiation at 45°S between October-February. (Please see the new Figure 4). We also expanded the discussion section in general.

Conclusions: The statement "Potassium was analyzed in order to provide a more complete biomass burning record through the comparison with another fire proxy" should be reworded. I would suggest instead "The comparison between levoglucosan and the multi-sourced Kbb suggests that Kbb is best used in conjunction with other biomass

burning markers."

Technical/minor comments:

Page 5, line 16: This would be appropriate in the background.

Reply: We moved this phrase to the background information on page 2.

While changes in the westerly wind belt would have changed climate in Patagonia, is it possible that changes in the regional scale winds would also have impacted transport of Levoglucosan to Antarctica?

Reply: We now include the following sentence in Section 3.3.: "This interpretation is consistent with the increase of more humid conditions in continental records ((Iglesias and Whitlock, 2014 and Iglesias et al. 2014) although it cannot be excluded that the changes in the westerly wind belt may have affected, at least in part, increased transport of aerosols to Antarctica."

Why is the levoglucosan record compared to Southern Hemisphere charcoal (page 10 line 27-28) when in section 3.3 the source was determined to be most likely Patagonia? P. 11 lines 10-12: this sentence is unclear.

Reply: The Southern Hemisphere compilation helps determine the relative contributions of both southern South America, we well as including the all location south of 45°S that do not include sufficient charcoal site to be individually discussed. We expanded this section, which we hope will show the use of the Southern Hemisphere compilation:

"Talos Dome Levoglucosan and southern South America charcoal records (Fig.3) both depict an increasing long-term trend in biomass burning from 4000 to 750 yr BP, although these trends do not perfectly coincide. These records differ both before and after these ∼3000 years of similarity. The Talos Dome record begins its major and sustained increase in fire activity around 4500 yr BP, while the southern South American record is decreasing during this time period. The southern South America and East

Australian records substantially diverge after 750 yr BP, where the southern South American fires decrease and the East Australian and SH fires both markedly increase. The Talos Dome levoglucosan record does not extend to the present, and so comparisons with this major difference in regional fire sources is not possible."

P.11 line 16: "This 1000-year peak in levoglucosan. . ." I would encourage the authors to define the age range of the peak referenced here. A reference is needed to support the statement "the cooling period coincides with a shift to drier conditions in South America (page 12 line 6-7)".

Reply: The sentence is now the following: "This 1000-year peak in levoglucosan (between 2450±100 and 1600±1000 yr BP) occurs at the same time as a local decrease in temperature between 1500 and 2500 yr BP."

Figure 2: The Y axis for plot 2B does not have a complete label. The caption in missing a parentheses.

Reply: We completely changed Figure 2, following both these suggestions as well as your earlier idea.

Figure 4: This plot would be clearer if the y-axes were labeled rather than having the labels floating within the plot. Does the 250 yr average window include the spikes? This is unclear since the spikes are plotted.

Reply: We changed Figure 4 to include axes that are easier to read. We also changed the plotted insolation We also agree that using from 30°S to 45°S between October-February.

---

## Author Comment (AC3) · 22 Mar 2018

Anonymous Referee #1 This manuscript reports on levoglucosan and the calculated fine (i.e., non sea-salt and non dust) potassium fraction measured on sub-samples from a Talos Dome (Antarctica) ice core spanning the second part of the Holocene. The aim of the work is to reconstruct past fire activity in the southern hemisphere. The data presented in the manuscript possibly contain information in view to better understand past variability of fire activity. This topic is clearly relevant for Climate of the Past journal. As it stands the manuscript, however, requires major revisions on several key aspects and a re-evaluation prior to publication. Reply: Thanks for your review and your observations that are surely useful for improving the quality of the paper. We substantially revised and rewrite the paper following your indications. We reply to specific

comments below. RC1 First, the wording "We reconstructed the high latitude Southern Hemisphere fire history by using the specific biomarker levoglucosan and potassium in ice cores from the TALDICE during the Mid-Late Holocene (750-6000 yr BP) is clearly an overstatement since your data are discontinuous (only 15% of the ice core was measured). Please, make this point more clear in the manuscript and change the wording in your discussion. Reply: Surely our data are not continuous. As you underlined, we analyzed the uppermost part (15 cm) of each 1 m section as reported in the experimental section. In this view some fire peaks can be missed. We tried to be more precise in this sense adding a sentence (P.4 L.7 of the revised version), where we stated that: "The samples therefore have the potential to miss fire peaks due to their discontinuous nature". RC1 Second, I would like to see in the manuscript the raw data (concentrations and not only calculated deposition flux) for levoglucosan and potassium. If my estimate is correct, you calculated fine potassium concentrations in the order of 0.2 ppb. I strongly recommend to the authors to show concentrations of potassium, sodium, and iron and error propagation estimates. Only with that the reviewers (and the readers) can evaluate to robustness of the potassium approach. Reply: In our view, adding concentration values in the manuscript may potentially burden the paper. To this purpose, we opted to include the concentration plots in supporting material and we'll add a raw data file (rawdata.xls), where original values (concentration) are reported for the readers and the reviewers. Error propagation was estimated using propagation error formula (that will be reported in Supporting Material). We noticed also a typo in the graph, where Kbb flux was reported in $\mu$g m-2 y-1, instead of mg m-2 y-1, thus probably arising your doubt about original concentration values. Fig. 2 strongly changed, also following the suggestions of the reviewer #2 and we corrected the typo in the units. RC1: Also concerning the discrepancies between levoglucosan and potassium: line 8 page 7: what other sources for potassium in the pre-industrial atmosphere (apart from sea-salt and dust) you have in mind here ???? Reply: Well, we don't have in mind other sources of potassium in the pre-industrial atmosphere. What we wanted to underline is that when disentangling the different contributions, you make an assumption

(i.e. terrestrial and marine composition is constant). Terrestrial sources, in particular, can change, where local or regional sources may affect your assumed composition and, in turn, your record, leading to anomalous values. We tried to clarify this point at P. 7 L. 21 in the revised version where we stated that "Differences [between Kbb and levoglucosan] may be due to the non-specificity of potassium for biomass burning where local to regional terrestrial sources may influence the trend, leading to differences with the levoglucosan signal". RC1 Third, you have ignored the study of black carbon from Antarctic ice covering the Holocene from Arienzo et al. (2017) published in the JGR in 2017. Even though the ice core discussed in this recent paper is from another Antarctic site, it is needed to refer to this previous work and discuss similarity and difference with your work. Note that this JGR paper is based on a continuous back carbon record. I attached below the abstract, showing how deep was the discussion of this record in terms of climatic condition changes in the south hemisphere and particularly in South America. Reply: We developed our discussion also considering the Arienzo work. It will be found in the section currently named "drivers for biomass burning", as suggested by RC#2. The paper that you indicated is surely interesting and it must be mentioned. In the Arienzo paper a correlation between black carbon (BC) and solar radiation is proposed, where a difference between October/February insolation is used. Considering that we hypothesized that Patagonia is the main source of levoglucosan signal in Talos Dome (based on charcoal records comparisons), we opted to compare the interval (October-February) in order to take into account of the major influence of the growing season (the driving force for North Patagonia fires) and the burning season (from December to February) that mainly drives South Patagonia fires. We opted for discussing the BC record reported by Arienzo et al., but we also are aware that the Antarctic site reported in this paper are quite distant from Talos Dome (as you recognized) and the BC record in the interval between 6000 and 2000 yr BP showed only few values, in contrast with the high resolution during the Early Holocene and the last two millennia. The few BC data in the Arienzo et al. paper doesn't allow to properly use a robust statistical approach to determine correlations between BC and

our levoglucosan record.

---

## Author Comment (AC4) · 22 Mar 2018

Dear Editor, find attached the revised version of the paper including supporting material and rawdata.xls. Best regards Dario

Please also note the supplement to this comment:
https://www.clim-past-discuss.net/cp-2017-158/cp-2017-158-AC4-supplement.zip

---

## Referee Report (RR1)

The answers and data provided by authors raise more questions than answers concerning potassium data, in particular. So sorry but your potassium data are likely incorrect.

In my first review, I asked to see in the manuscript the raw data (concentrations and not only calculated deposition flux) for levoglucosan and potassium. Indeed, if my estimate was correct, based on plots reported in the first version, I calculated fine potassium peak concentrations in the order of 0.2 ppb. I thus strongly recommend to the authors to show concentrations of potassium, sodium, and iron and error propagation estimates to evaluate the robustness of the potassium approach present at such low levels. In your response you indicate that you made a unit error and in fact potassium peaks reach 200 ppb.

I am strongly surprised by such high potassium values since many studies conducted in Greenland have shown that, if attributable to biomass burning, the fine potassium perturbations never exceed a few ppb. Also emission factors of potassium and fine potassium from biomass burning are in the same magnitude (Akagi et al., Andreae and Merlet, Gao et al) and if we assume a similar lifetime for the two species we may expect input of similar amplitude for fine potassium and levo. The difference between Kbb and levo in our data suggests a far shorter atmospheric lifetime for levo than for fine potassium !

So checking your chemistry Excel table, I report below two figures. Your sampling is 15 cm and I also report in Fig 2 the sodium profile reported from Schüpbach et al. (10 cm resolution). We can see that you have several sodium peaks exceeding 200 ppb (up to 1.3 ppm, Fig.1) and it is less frequent in the continuous profile from Schüpbach. Even more surprising for me, when I discover that your sodium peaks coincide with potassium ones (Figure 3). Why did you totally miss to comment that in the manuscript: it is a critical point (see below).

I scrutinize the plots of potassium versus sodium (Fig 3A) and also selecting samples with low sodium levels (<100 ppb) (Fig 3B). Even here you quasi never reach the seawater ratio (blue line in Fig 3B). That is surprising for antarctic ice. If I refer to Fe or Ca there is no way to explain that with terrestrial potassium (Fig 3C).

In conclusion, I don't see any issue to calculate fine potassium with your data (even as an estimate since your measurements indicate that the sea-salt potassium to sodium ratio is quasi never reached).

Finally, just a comment outside the potassium topic: I don't think that in your Fig 2A is accumulation rate: I am not sure but I think you reported ice annual thickness ??? Anyway, that does not fit with the plots of Schüpbach et al. (see figure 2 below).

[Figure]

*Figure 1 (your data)*

[Figure]

*From Schüpbach et al 2013 (Figure 3)*

[Figure]

*Figure 3 (your data)*

---

## Author Response (AR2)

**Reply to Editor**

Thank you for the opportunity to revise our paper. After a careful re-evaluation of our data in conjunction with the comments of the referees, we recognized several problems in the potassium record. We especially thank the reviewer #1 for his/her thorough evaluation that underlined this point. We agree that potassium values need to be re-evaluated to the point of being completely reanalyzed analytically. As removing the potassium record does not actually affect the following discussion in the paper, we now completely remove the discussion regarding potassium as a biomass burning marker in this work and we now only focus on the levoglucosan record.
* * *
**Reply to Referee #1.**

Thank you for your review. We appreciate your thorough evaluation of potassium data and we agree with the discrepancy that you underlined. We realize that many reasons may have led to this discrepancy, including instrumental or calibration issues. We therefore opted to not include the potassium record in our paper, as it does not affect the discussion and interpretations.

Referring to your comment : "We can see that you have several sodium peaks exceeding 200 ppb (up to 1.3 ppm, Fig.1) and it is less frequent in the continuous profile from Schüpbach", we would like to mention that the comparison between Schüpbach et al.'s work and our Na record is limited to the depth range between 425 and 300m. In this interval, sodium peaks do not exceed 200 ppb, which differs from the comments in the review. Please note that Schüpbach et al. plot their data using a log-scale that may influence a direct visual comparison.

The final comment regarding possible problems in the accumulation rate values reported in Fig. 2A may be due to the fact that Schüpbach et al's Fig. 3 covers a depth range of 1400-300 m, while our record covers 425-90 m.
* * *
**Reply to Referee #2.**

Thank you for your suggestions.

Major:

Page 7 line 35. We incorporated the comments from Reviewer #1 and the editor, we have now removed all discussions of the potassium record.

Page 7 line 35 to page 8 line 9. We agree with your point regarding that individual spikes may also be due to other possibilities represent We delete large part of the discussion about spikes. We have always agreed with your point about individual spikes as they are not likely representative of megafires. This part is removed and we leaved only a small paragraph (p8 line 21-35) that briefly reported about this evidence.

Section 3.2. We add a sentence (page 9 line. 23-24 and a new reference –Antony et al. 2014) to improve this discussion

Minor:

General: we changed all the time setting from oldest to youngest along the paper

Page 7 line 4. We reword the introductory sentence (Page 7 line 5).

Page.8 line 10. We deleted this paragraph (see ref#1 and editor reply).

Figure comments: we changed the captions following your indications.

[revised manuscript text omitted]